# Drug Utilization for Pain Management during Perioperative Period of Total Knee Arthroplasty in China: A Retrospective Research Using Real-World Data

**DOI:** 10.3390/medicina57050451

**Published:** 2021-05-06

**Authors:** Xianwen Chen, Lisong Yang, Xueli Liu, He Zhu, Fei Yu, Carolina Oi Lam Ung, Hao Hu, Waisin Chan, Honghao Shi, Sheng Han

**Affiliations:** 1State Key Laboratory of Quality Research in Chinese Medicine, Institute of Chinese Medical Sciences, University of Macao, Macao 999078, China; mb85843@connect.um.edu.mo (X.C.); mb85829@connect.um.edu.mo (L.Y.); carolinaung@um.edu.mo (C.O.L.U.); haohu@um.edu.mo (H.H.); 2International Research Center of Medicinal Administration, Peking University, Beijing 100080, China; shirley.liu@pkuircma.org.cn (X.L.); kevin.zhu@pkuircma.org.cn (H.Z.); 3Peking University First Hospital, Peking University, Beijing 100032, China; faye.yufei@gmail.com; 4Orthopaedic Department, Conde S. Januario General Hospital, Macao 999078, China; drchanws@gmail.com

**Keywords:** total knee arthroplasty, perioperative, analgesics, pain management, drug utilization, surgery, anesthesia

## Abstract

*Background and Objective:* Total knee arthroplasty (TKA) is one of the most painful procedures and perioperative pain usually requires the use of many analgesics to relieve it. The appropriate use of analgesics to relieve patient pain is an important issue of TKA. To characterize the drug utilization for pain management during perioperative period of TKA in China using real-world data of electronic medical records. *Materials and Methods:* This research used the data of all inpatients who received TKA at 145 hospitals covered 31 provinces in China from 1 January 2016 to 31 December 2018. The exclusion criteria included pregnancy and cancer diagnosis. In the analysis of drug utilization mode (DUM), medicines were classified into 5 groups: non-steroidal anti-inflammatory drugs (NSAIDs), opioids, non-opioid central analgesics, acetaminophen and others. *Results:* Among the 2017 patients included in this study, there were 1537 (76.20%) female and 480 (23.80%) male, aged 65.77 ± 7.73 years. Regarding the surgery characteristics, 1658 (82.20%) were unilateral; 1220 (60.49%) was graded Level 4; 1312 (65.05%) used local anesthesia as the main anesthesia method, and 1450 (71.89%) lasted for more than 2 h. The most common DUM was “NSAIDs + opioids” (55.92%), followed by “NSAIDs only” (17.85%), and “NSAIDs + Opioids + Non-opioid central analgesics” (17.15%). The results of the Chi-square test showed that differences in DUM were associated with surgery types, surgery levels, surgery duration, and types of anesthesia used. Up to 81.14% of the total drug expenses for pain management was spent on NSAIDs. Due to the limitation of database, this study could not subdivide operation stages, anesthesia methods, dosage forms of drugs. *Conclusion:* In China, the use of analgesics in perioperative period of TKA was diversified and influenced by a number of surgery characteristics. The rational use of analgesics should be considered in combination with surgery type, surgery level, surgery duration and anesthesia method.

## 1. Introduction

Total knee replacement (TKA) refers to a treatment that removes articular surfaces which cannot be repaired by itself and replaces them with artificial joints to eliminate pain and stabilize the knee joint, and can be unilateral for one-sided lesion or bilateral for two-sided lesion [1,2]. In 2012, according to a survey in 18 countries, about 1.1 million people received TKA every year, and the number increased by 11% yearly [3]. In 2016, there were nearly 110 million knee osteoarthritis patients in China, and the potential patient population requiring TKA is enormous [4].

TKA is one of the most painful procedures surgical procedures [5,6]. Appropriate pain management is one of the utmost concerns for patients receiving TKA. Postoperative pain of TKA is one of the important factors affecting patients’ rehabilitation [7]. Improper pain control can affect the normal recovery of knee function, cause chronic pain, reduce the quality of life, and increase the risk of complications [8,9,10,11]. Furthermore, severe pain can also increase stress and cause sleep disorders, which may in turn reduce the patient’s immunity and increase the risk of infection [12,13,14,15]. Long-term postoperative pain may also lead to the increased use of opioids [16], prolong the hospital stay length, and increase individual medical costs and the overall social burden [17,18].

At present, no consensus has been reached about the use of analgesics for pain management in TKA. The American Academy of Pain Medicine (AAPM), the American Society of Anesthesiologists (ASA) and other professional organizations suggested that multimodal analgesics may be used for pain management in TKA and the recommended medicines included acetaminophen, non-steroidal anti-inflammatory drugs (NSAIDs), gabapentin pregabalin and ketamine, etc. [10]. The Korean Knee Society recommended that two or more medicines having different mechanisms of action and acting on different body parts should be used for TKA patients [11]. Research gaps in establishing a standardized evidence-based practice guidelines for operative pain management remains [19].

In practice, multimodal analgesia has been recommended as an alternative consid-ering the risks of opioid monotherapy in surgical pain management [20]. Clinicians have long been aware of the danger of overreliance on opioids to manage acute pain: the higher risks of adverse drug events, increased length of stay and hospital costs, and decreased patient satisfaction [21,22]. Multimodal analgesia refers to the approach that relies on a nonopioid foundation with addition of adjunctive opioids as needed [23]. An increasing number of nonopioid analgesics such as NSAIDs, acetaminophen and tramadol have been proved effective for this role, with fewer side effects and a higher degree of safety than opioids [16,17]. Accordingly, multimodal analgesia was recommended as best practice by most recognized authorities [13,15,24,25]. 

Previous studies have reported the use of multimodal analgesia in TKA cases. For example, Lubis AMT et al. analyzed the combined use of NSAIDs after TKA, showing that celecoxib combined with pregabalin can effectively relieve pain while Drew JM et al. recommended celecoxib and oxycodone for TKA patients [26,27]. Smith SR et al. studied the use of opioids in patients before TKA, and found that patients who used opioids preoperatively suffered more pain [28]. Currently, the studies about the use of multimodal analgesia were mainly TKA cases in western countries and research about such use among Asian patients was very limited.

Therefore, the aim of this research was to characterize the drug utilization for pain management during perioperative period of TKA in China by using real world data, and to analyze the association between drug utilization and perioperative characteristics. The results of this study were expected to provide a reference for the future development of guidelines for the use of perioperative analgesics in TKA.

## 2. Materials and Methods

### 2.1. Research Design and Data Collection

The research applied a retrospective observational research design using data extracted from the electronic medical records of hospitals. The research was reviewed and approved by the Ethics Committee of Peking University for the project of National Natural Science Foundation of China (71603008). The ethical code: IRB00001051-19034, approved date: 25 April 2019.

All the data was collected from 145 hospitals in 31 provinces across China, dated between 1 January 2016 and 31 December 2018. The electronic medical records without personal identity information were directly extracted from the Health Information Systems of the hospitals. 

### 2.2. Sample Inclusion and Exclusion

The inclusion criteria of this study were hospitalized patients who: (1) aged 18 years or older; (2) were admitted and discharged during the study period; and (3) only underwent TKA (including unilateral and bilateral) but no other surgery; and (4) had a record of analgesics prescriptions. The exclusion criteria were: (1) pregnancy during hospitalization; (2) diagnosis of cancer during hospitalization; (3) underwent operations in addition to TKA; (4) no record of analgesic use; and (5) incomplete patient information.

### 2.3. Surgery Characteristics

The characteristics of surgery were measured by four aspects:The types of surgery were divided into unilateral surgery and bilateral surgery.Surgery were classified based on different risks and complexity of surgery in accordance with Article 7 of Chapter II of the 2012 edition of the “Measures for the Management of Surgery in Medical Institutions (Trial)” [29] into four levels: Level 1 refers to surgery with lower risk, simple procedure, and low technical difficulty; Level 2 refers to surgery with certain risks, moderate process complexity, and technical difficulty; Level 3 refers to higher-risk, more complicated, and more difficult surgery; and Level 4 refers to higher-risk, complicated, and difficult surgery.Anesthesia methods were divided into local anesthesia and general anesthesia.The surgery duration was calculated from the beginning to the end of the TKA operation.

### 2.4. Analgesics

The analgesics in this study were named using the generic names and classified using the classification system of drugs used as perioperative analgesia employed in the “Experts consensus on postoperative pain management in adults (2017)” and “Experts consensus on perioperative pain management in general surgery (2015)” [30,31]. Tramadol was classified as a non-opioid central analgesic rather than a weak opioid according to expert consensus in this study. They included:(1)Opioids: Morphine, Oxycodone, Hydromorphone, Sufentanil, Hydrocodone, Fentanyl, Butorphanol, Dezocine, Pethidine, Pentazocine, Nalbuphine, Buprenorphine, Codeine, Dihydrocodeine(2)Non-steroidal anti-inflammatory drugs (NSAIDs): Ibuprofen, Diclofenac, Meloxicam, Celecoxib, Lornoxicam(3)Acetaminophen(4)Non-opioid central analgesics: Tramadol(5)Others: Ketamine, D-ketamine, Gabapentin, Pregabalin

In this study, analgesic mode referred to the combination and type of analgesics used by a patient during a perioperative TKA in this study.

### 2.5. Data Analysis

The study used descriptive statistics to analyze the characteristics of TKA patients, surgery characteristics, and costs. Chi-Square Test was used to analyze the relationships among surgery characteristics and analgesic modes in TKA patients. SPSS 24.0 software (IBM corporation, Armonk, NY, USA) was used. 

All analyses were two-tailed, and *p* < 0.05 was considered as statistics significance.

## 3. Results

### 3.1. Background Information

#### 3.1.1. Characteristics of Patients

A total of 2017 patients (surgeries) were included in this study. As shown in Table 1, the mean age of the patients was 65.77 ± 7.73 years old, with 1334 (66.14%) of them aged 51–70 years; 1537 (76.20%) were female and 480 (23.80%) were male; 2002 (99.26%) patients had reimbursement support, and 1880 (93.21%) received treatment in tertiary hospital. The average length of hospitalization was 14.35 days, and 1031 (51.12%) patients stayed for more than 14 days in the hospital.

#### 3.1.2. Surgical Characteristics

With regards to the characteristics of the surgery, as shown in Table 2, 1658 (82.20%) were unilateral, 1220 (60.49%) were graded Level 4 and 750 (37.18%) were Level 3; 1312 (65.05%) used local anesthesia; and 1450 (71.89%) lasted for more than two hours. The mean duration of unilateral was 2.55 ± 1.07 h, while the mean duration of bilateral was 3.98 ± 1.60 h.

#### 3.1.3. Cost Information

The patients’ costs were shown in Table 3. The cost was calculated from the time the patient was hospitalized and ended when the patient was discharged. The top three costs were treatment (50.85%), operation (18.14%) and medicine (9.07%). The cost of analgesics accounted for 1.40% of the total hospitalization cost and accounted for 15.41% of all drug costs.

### 3.2. Analgesics Utilization

#### 3.2.1. Modes of Analgesics Utilization

The most common analgesic drugs used were NSAIDs, followed by opioids and non-opioid central analgesics. Most TKA patients used more than 1 analgesics from different drug groups for pain management. As shown in Table 4, the TOP 5 modes of analgesics utilization were: “NSAIDs + opioids” (55.92%), “NSAIDs” (17.85%), “NSAIDs + opioids + non-opiates central analgesics” (17.15%), “Opioids” (4.96%), and “NSAIDs + non-opioid central analgesics” (2.78%). 

#### 3.2.2. Analgesics Used in the Most Common Utilization Modes

Analgesics were mostly used in combination, and the way of combined use varied even in a mode. The top 3 drug combinations in the top three modes of analgesics utilization were shown in Table 5. 

NSAIDs drugs were the most commonly used in the combination of drug use. The major drugs were Celecoxib, Parecoxib and Flurbiprofen axetil. The most common opioids used were Morphine and Sufentanil.

### 3.3. Association between the Modes of Analgesics Utilization and the Surgery Characteristics

The analysis results of the association between the modes of analgesics utilization and the surgery characteristics were shown in Table 6. Surgery types, surgery level, anesthesia methods and surgery duration were associated with the mode of analgesics utilization with statistical differences.

### 3.4. Cost Analysis of Analgesics

Most of the costs on analgesics were spent on NSAIDs, which cost the most analgesics use totaling 1579,037.55 CNY (83% of the total analgesic expenses), followed by opioids totaling 312,279.05 CNY (16% of the total analgesic expenses), non-opioid central analgesics totaling 22,027.57 CNY (1% of the total analgesic expenses) and other analgesic categories totaling 4125.22 CNY (<1% of the total analgesic expenses).

Among the analgesic drugs used by TKA patients, there were 7 NSAIDs, 6 opioids, 1 non-opioid central analgesics, and acetaminophen. In terms of cost, the top 5 drugs were parecoxib, flurbiprofen axetil, sufentanil, celecoxib and remifentanil, making up more than 80% of the total analgesic expenses (as shown in Table 7).

## 4. Discussion

In this retrospective study, it was found that multimodal analgesic was commonly prescribed for more than 95% of the patients in this study which was similar to findings from previous studies about postoperative pain management [8]. NSAIDs including parecoxib, flurbiprofen axetil and celecoxib were most commonly prescribed, which was also in accordance with previous findings [17]. In previous studies, tramadol was classified as a weak opioid agonist with two mechanisms of action [32]. And in some study, tramadol was identified as an atypical opioid distinct from opioids [33]. In this study, tramadol was classified as a non-opioid central analgesic based on existing expert consensus in China, which is a reflection of real-world drug utilization method in China and was therefore adopted. Surgery types, surgery levels, surgery duration, and types of anesthesia were the surgery characteristics found to have associations with the mode of drug utilization with statistical significance, which indicated that the surgery attributes should be taken in consideration when developing analgesics guideline in the future.

In this research, different modes of drug utilization used for pain management in TKA were identified. Considering the lack of guideline and quality evidence, such use could mostly be at the discretion of the prescriber based on his/her professional and clinical knowledge and experiences. These findings highlighted the lack of standardization of pain management [34]. For instance, “celecoxib + parecoxib” was the most common mode of combined drug use, and 29 patients used Sufentanil and Remifentanil at the same time in Mode 3, both of which had the same mechanism of action. Although, there is some evidence showed that if iv parecoxib followed by oral celecoxib in different period after TKA surgery may decrease the use of morphine [35]. However, in a study analyzing international reports of adverse reactions, celecoxib was found to pose a greater risk of adverse reactions when used in combination with other NSAIDs, such as gastrointestinal side effects, renal and urinary disorders, and hypertension [36]. One study also confirmed a higher risk of gastrointestinal bleeding with the combination of different NSAIDs [37]. So, it is still necessary to be vigilant when using drugs with the same mechanism of action during perioperative period and needs further exploration [38]. Overall, the lack of quality evidence of rational analgesic use for pain management in TKA demands efforts from researchers.

This research also found that the main analgesics used by TKA patients for pain management were celecoxib, parecoxib, flurbiprofen axetil, morphine, tramadol, and sufentanil, etc. but not acetaminophen. Acetaminophen, as an analgesic, is often used to assist the analgesic effect of other drugs. Murata-ooiwa M et al. [18] and Mont Ma et al. [39] both confirmed the effect of acetaminophen on the pain management after TKA surgeryin achieving effective pain control for the patients. Mont Ma et al. [39] had also confirmed that the use of acetaminophen could reduce the probability of reoperation and save the cost of patients. In this study, very few TKA patients used acetaminophen. This might be due to the consideration that acetaminophen does not have anti-inflammatory activity but liver toxicity. Other types of analgesics, such as pregabalin was also used by the patients in this study. Some research results showed that while this other type of analgesics such as gabapentin and pregabalin could alleviate postoperative pain and reduce the use of opioids [39,40], the adverse reactions and effectiveness of the drugs have yet to be confirmed [41,42,43,44]. How to avoid adverse reactions of analgesics in TKA needs more clinical research.

Overall, the pain management of perioperative in China still needs improvement and refinement. Previous retrospective studies have confirmed the existence of various irrational phenomena in the current use of orthopaedic analgesics in China, such as repeated use of drugs and unreasonable dosage [45,46]. The rational use of perioperative analgesics (especially in orthopaedics) should be expedited by formulating guidelines and further optimising management.

To our knowledge, this is the first research to report the real-world utilization of analgesic drugs for pain management during perioperative period of TKA. However, there were some limitations in this research and needs further study. Firstly, due to the limitation of database, this study only analyzed the whole perioperative period, and did not distinguish the different stages of operation (preoperative, intraoperative, or postoperative) or different anesthesia methods (pure local anesthesia, regional anesthesia or neuraxial anesthesia). Secondly, this study did not specify the dosage forms of analgesic drugs. This is relevant because different dosage forms may correspond to use at different times. In the follow-up study, it needs to combine the time and dosage form of the drug, refine the mode of drug utilization for pain management, and use the patient’s pain score and other clinical indicators to evaluate the effect of the utilization modes of analgesics.

## 5. Conclusions

In conclusion, there were a variety of perioperative analgesics utilization modes for pain management of TKA patients. The decision about analgesic use might need to take into consideration the surgery type, surgery level, surgery duration, and anesthesia method. Further clinical evidence is needed to support the development of guidelines for perioperative analgesic drug use of TKA.

## Figures and Tables

**Table 1 medicina-57-00451-t001:** Basic characteristics of TKA patients (*n* = 2107).

Variable	TKA (*n* = 2017)
Age (mean ± SD, yr)	65.77 ± 7.73
Age group (*n*, %)	
≤50	48 (2.38%)
51–70	1334 (66.14%)
≥71	635 (31.48%)
Gender (*n*, %)	
Female	1537 (76.20%)
Male	480 (23.80%)
Reimbursement (*n*, %)	
Yes	2002 (99.26%)
No	15 (0.74%)
Hospital level	
Tertiary	1880 (93.21%)
Others	137 (6.79%)
Length of hospitalization (D)	14.35 ± 6.27
<14	986 (48.88%)
≥14	1031 (51.12%)

Abbreviations: TKA: Total Knee Arthroplasty; SD: Standard Deviation; D: Days.

**Table 2 medicina-57-00451-t002:** Surgical characteristics of TKA patients (*n* = 2107).

*n* = 2017	Number of Operations (*n*)	Proportion (%)
Surgery types		
Unilateral	1658	82.20%
Bilateral	359	17.80%
Surgery level		
Level 4	1220	60.49%
Level 3	750	37.18%
Level 2	31	1.54%
Level 1	16	0.79%
Anesthesia method		
Local anesthesia	1312	65.05%
General anesthesia	705	34.95%
Surgery duration (h)		
Overall	2.81 ± 1.30	
<2	567	28.11%
≥2	1450	71.89%
Unilateral	2.55 ± 1.07	
<2	557	33.59%
≥2	1100	66.34%
Bilateral	3.98 ± 1.60	
<2	10	2.79%
≥2	349	97.21%

**Table 3 medicina-57-00451-t003:** Costs composition of TKA patients (*n* = 2107).

Cost Types	Average (CNY)	Standard Deviation	Proportion
Treatment	34,606.34	30,520.53	50.85%
Operation	12,346.49	16,781.28	18.14%
Medicine	6169.216	4998.09	9.07%
Analgesics	950.65	828.49	1.40% (15.41%) *
Laboratory	2123.03	985.62	3.12%
Examination	1880.84	944.95	2.76%
Bed	863.28	696.11	1.27%
Nursing care	412.33	295.16	0.61%
Other	9651.20	19,589.36	14.18%
Total	68,052.73	23,743.28	100.00%

* It refers to the proportion of the analgesics in the medicine expenses.

**Table 4 medicina-57-00451-t004:** Top 5 modes of analgesics utilization based on drug groups.

Top 5 Modes	*n* *	Proportion
Mode 1: Nonsteroidal Antiinflammatory Drugs (NSAIDs) + Opioids	1128	55.92%
Mode 2: NSAIDs only	360	17.85%
Mode 3: NSAIDs + Opioids + Non-opioid central analgesics	346	17.15%
Mode 4: Opioids only	100	4.96%
Mode 5: NSAIDs + Non-opioid central analgesics	56	2.78%

*n* *: The number of prescriptions. Abbreviations: NSAIDs: Nonsteroidal Antiinflammatory Drugs.

**Table 5 medicina-57-00451-t005:** Top 3 drug combinations in main modes of analgesics utilization.

Mode Type	Drug Combinations	*n*	Proportion *
Mode 2 (*n* = 360)	Celecoxib + Parecoxib	79	21.94%
	Celecoxib + Flurbiprofen axetil	50	13.89%
	Flurbiprofen axetil	40	11.11%
Mode 1 (*n* = 1128)	Tromethamine + Sufentanil	84	7.45%
	Celecoxib + Parecoxib + Morphine	72	6.38%
	Flurbiprofen axetil + Sufentanil	58	5.14%
Mode 3 (*n* = 346)	Celecoxib + Parecoxib + Morphine + Tramadol	39	11.27%
	Celecoxib + Parecoxib + Morphine + Tramadol + Flurbiprofen axetil	34	9.83%
	Celecoxib + Flurbiprofen axetil + Sufentanil + Remifentanil + Tramadol	29	8.38%

* The proportion in one mode, for example: 21.94% means 79/360 × 100%.

**Table 6 medicina-57-00451-t006:** Analysis of factors influencing the main mode of analgesic.

Factors	Mode 1 (*n* = 360)	Mode 2 (*n* = 1128)	Mode 3 (*n* = 346)	Sum	χ^2^	*p*-Value
Surgery types						
Unilateral	285 (19.03%)	938 (62.62%)	275 (18.36%)	1498 (100.00%)	9.623	0.008
Bilateral	75 (22.32%)	190 (56.55%)	71 (21.13%)	336 (100.00%)		
Surgery level						
Level 4	229 (20.11%)	677 (59.44%)	233 (20.46%)	1139 (100.00%)	16.718	0.010
Level 3	128 (19.60%)	415 (63.55%)	110 (16.85%)	653 (100.00%)		
Level 2	3 (11.11%)	21 (77.78%)	3 (11.11%)	27 (100.00%)		
Level 1	0 (0.0%)	15 (100.00%)	0 (0.0%)	15 (100.00%)		
Anesthesia method						
Local anesthesia	304 (25.59%)	671 (56.84%)	213 (17.93%)	1188 (100.00%)	76.44	0.000
General anesthesia	56 (8.67%)	457 (70.74%)	133 (20.59%)	646 (100.00%)		
Surgery duration (h)						
≥2	229 (17.31%)	807 (61.00%)	287 (21.69%)	1323 (100.00%)	16.602	0.000
<2	117 (22.90%)	321 (62.82%)	73 (14.29%)	511 (100.00%)		

**Table 7 medicina-57-00451-t007:** Analgesic drug cost.

Generic Name	Expenses (CNY)	Proportion
Parecoxib	749,130.60	39.07%
Flurbiprofen axetil	576,556.60	30.07%
Sufentanil	129,385.70	6.75%
Celecoxib	115,159.90	6.01%
Remifentanil	102,064.50	5.32%
Tromethamine	81,548.44	4.25%
Oxycodone	72,091.81	3.76%
Diclofenac	43,699.63	2.28%
Tramadol	22,027.57	1.15%
Lornoxicam	12,397.83	0.65%
Morphine	7798.16	0.41%
Pregabalin	4125.22	0.22%
Fentanyl	811.96	0.04%
Ibuprofen	480.00	0.03%
Dezocin	127.00	0.01%
Acetaminophen	64.52	0.00%

## Data Availability

The data used to support the findings of this study are available from the corresponding author upon request.

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
