# Peer review of "Drug Utilization for Pain Management during Perioperative Period of Total Knee Arthroplasty in China: A Retrospective Research Using Real-World Data"

_medicina, 2021, doi:10.3390/medicina57050451_

Round 1

Reviewer 1 Report

  1. I am quite suprised that a Tramadol is named non-opioid drug. It is for sure atypical centrally-acting drug, but (+)-Tramadol and metabolite (+)-O-desmethyl-tramadol (M1) are agonists of the mu opioid receptor.
  2. ..."The aim of this research was to characterize the drug utilization for pain management during perioperative period of TKA in China by using real world data, and to analyze the association between drug utilization and perioperative characteristics. The results of this study are expected to provide a reference for the future development of guidelines for the use of perioperative analgesics in TKA."...- what the point is in showing what drugs are used without any correlations with efficacy? Are there any data for VAS/NRS scores pre and post TKA? How the data presentad will influence a future development of the guidelines?
  3. The different drug utilization modes are showed in the manuscript- mostly combined. But there is no information how the drugs were administered in a time manner- all together? one after one?. There is a suggestion in one or two sentences, that in some combinations the drugs were administered simultaneously (lines 211, 216). In the situation of eg. celecoxib+parecoxib it seems like medical mistake- same mechanism of action, no additional analgesic effect, much more potential adverse effects. In the work of Zhuang Q et al there is nothing, which can support such a "same time combination" (Reference 31)- firstly it is just description of the protocol of the study to be done, secondly in the protocol iv parecoxib will be followed by oral celecoxib (not simultaneously).

Author Response

Responses to reviewer 1:

  1. I am quite surprised that a Tramadol is named non-opioid drug. It is for sure atypical centrally-acting drug, but (+)-Tramadol and metabolite (+)-O-desmethyl-tramadol (M1) are agonists of the mu opioid receptor.

Reply 1: Thank you for this comment. Firstly, in China, doctors mostly refer to the official expert consensus in the corresponding disease area for the way they use drugs. We have therefore selected two Chinese expert consensuses in order to find a classification that is consistent with the current state of analgesic use in China, which are Expert consensus on postoperative pain management in adults (2017) and Consensus of Experts in Pain Management in General Surgery (2015) (line 121-124). Both expert consensuses distinguish Tramadol from opioids. And in the second expert consensus, Tramadol is explicitly classified as a non-opioid central analgesic. Therefore, we considered this to be one such classification as a way of distinguishing Tramadol from opioids for central analgesia, and so it was chosen for the study methodology.

  1. "The aim of this research was to characterize the drug utilization for pain management during perioperative period of TKA in China by using real world data, and to analyze the association between drug utilization and perioperative characteristics. The results of this study are expected to provide a reference for the future development of guidelines for the use of perioperative analgesics in TKA."...- what the point is in showing what drugs are used without any correlations with efficacy? Are there any data for VAS/NRS scores pre and post TKA? How the data presented will influence a future development of the guidelines?

Reply 2: Thank you for this important comment.

Firstly, our study focused on a large sample of non-efficacy drug utilization data in real-world, such as drug amount, drug combination, etc., as a means of verifying whether analgesic utilization in China was consistent with guidelines or expert consensus, or whether there was irrational drug use. Then, we obtained results on the current state of drug use.

Secondly, our study was limited by the large sample size and the fact that the study was retrospective, which made it more difficult to design and carry out pain scoring research. We have included this in the research limitations section (line 248-256). “Firstly, due to the limits of database, this study only analyzed the whole perioperative period, and did not distinguish the different stages of operation (preoperative, intraoperative or postoperative) or different anesthesia method (pure local anesthesia, regional anesthesia or neuraxial anesthesia). Secondly, this study did not specify the dosage forms of analgesic drugs. This is relevant because different dosage forms may correspond to use at different times. In the follow-up study, it is needed to combine the time and dosage form of the drug, refine the mode of drug utilization for pain management, and use the patient's pain score and other clinical indicators to evaluate the effect of the utilization modes of analgesics.”

Thirdly, we hope to use the real analgesic use situation as reflected by the large sample data to identify the deficiencies and shortcomings in the current analgesic use situation, so as to provide reference for the guideline or expert consensus makers, with a view to improving the subsequent guideline formulation and further regulating the clinical use of drugs by doctors.

  1. The different drug utilization modes are showed in the manuscript- mostly combined. But there is no information how the drugs were administered in a time manner- all together? one after one? There is a suggestion in one or two sentences, that in some combinations the drugs were administered simultaneously (lines 211, 216). In the situation of eg. celecoxib+parecoxib it seems like medical mistake- same mechanism of action, no additional analgesic effect, much more potential adverse effects. In the work of Zhuang Q et al there is nothing, which can support such a "same time combination" (Reference 31)- firstly it is just description of the protocol of the study to be done, secondly in the protocol iv parecoxib will be followed by oral celecoxib (not simultaneously).

Reply 3: Thank for this insightful comment.

Firstly, due to the limitation of the data, the study cannot distinguish the time of medication, which we also interpreted it in the study limitations section (line 248-252). This study focused on the drug combination used in the perioperative period, and we analyzedthe drug combination without distinguishing the use nodes, mainly to analyze the overall situation of the utilization of analgesics in the perioperative period.

Secondly, we modified the referred of results of Zhuang Q et al to support “celecoxib + parecoxib” can be used in different timing after TKA. “Although, there is some evidence showed that if iv parecoxib followed by oral celecoxib in different period after TKA surgery may decrease the use of morphine, it is still necessary to be vigilant when using drugs with the same mechanism of action during perioperative period and needs further exploration.” (line219-223)

Thanks again for your professional comments and help!

Reviewer 2 Report

Major concerns : 

The authors have described the percentage of systemic analgesic used after TKAin a multimodal approach 

by definition multimodal approach should also consider regional anesthesia 

In this study the majority of patients had local anesthesia , this should be more precise was it pure local anesthesia , regional anesthesia or  neuraxail anesthesia 

regional techniques are described as being the most efficient technique in opoiod sparing effect 

This is a real major concern additional data are necessary about this topic 

Level of surgery should also be described in the table 

TKA , are one of the most painful surgical procedures this information should be mentionned and references  added 

The authors should provide a practical benefit for displaying these data right now I just don't see how these data could provide  clinical benefit 

I am not sure how to interpret the economic data what about cost of local anesthetics 

What the reimbursement aspect of these data bring to the reader ? 

In the discussion section , the mention of sufentanil and remifentanil used together is not clear to me what does it have to do with this current study if it has nothing to do , this paragraph should be deleted 

the results of these study should be compared to similar studies in this area and included in the conclusion , as the current  conclusion is not informative 

Author Response

Responses to reviewer 2:

Reviewer #2:

  1. The authors have described the percentage of systemic analgesic used after TKA in a multimodal approach by definition multimodal approach should also consider regional anesthesia. In this study the majority of patients had local anesthesia, this should be more precise was it pure local anesthesia, regional anesthesia or neuraxial anesthesia regional techniques are described as being the most efficient technique in opioid sparing effect. This is a real major concern additional data are necessary about this topic

Reply 1: Thank you for your important comment. Due to database limitations, we were not able to specifically break down the other categories of anesthesia. And we have added this point to our limitations section:

“Firstly, due to the limitation of database, this study only analyzed the whole perioperative period, and did not distinguish the different stages of operation (preoperative, intraoperative, or postoperative) or different anesthesia method (pure local anesthesia, regional an-esthesia or neuraxial anesthesia).”

  1. Level of surgery should also be described in the table.

Reply 2: Thank you for this comment. We have described level of surgery in Table 2 (line 158) to show the overall surgery type in our study and we used level of surgery as factors to analysis the influence for different analgesics model in Table 6 (line 191).

  1. TKA, are one of the most painful surgical procedures this information should be mentioned and references added.

Reply 3: Thank you for this comment. We have added this information in abstract and introduction (line15-17, 47).

  1. The authors should provide a practical benefit for displaying these data right now I just don't see how these data could provide clinical benefit. I am not sure how to interpret the economic data what about cost of local anesthetics. What the reimbursement aspect of these data brings to the reader?

Reply 4: We thank you for this important comment. Our study aimed to explore the use of real-world analgesic use data to inform policy or guideline makers. The results of the data showed that the current use of perioperative analgesia in China is concentrated and problematic, and therefore we need to publish more standardized guidelines to guide the rational use of analgesia in the clinical setting.

Secondly, due to the limitations of the database, we were unable to analyze the cost of anesthetic procedures.

Thirdly, the reimbursement data in this study were analyzed to reflect the importance of analgesics in health insurance. The fact that the vast majority of patients receive reimbursement for analgesics also demonstrates that emphasis should be placed on the rational use of analgesics, which may result in savings in health insurance costs.

  1. In the discussion section, the mention of sufentanil and remifentanil used together is not clear to me. What does it have to do with this current study, if it has nothing to do, this paragraph should be deleted. The results of these study should be compared to similar studies in this area and included in the conclusion, as the current conclusion is not informative.

Reply 8: Thank you for this comment. In our research, we have found the use of drugs with the same mechanism of action, such as “celecoxib and parecoxib” and “sufentanil and remifentanil”, which is likely to be the result of lack substandard medication practices or guidelines. And we have added more similar studies about the utilization of analgesics in China to enrich evidence of our findings and conclusion, such as “Overall, the pain management of perioperative in China still needs improvement and refinement. Previous retrospective studies have confirmed the existence of various irra-tional phenomena in the current use of orthopaedic analgesics in China, such as repeated use of drugs and unreasonable dosage [1, 2]. The rational use of perioperative analge-sics (especially in orthopaedics) should be expedited by formulating guidelines and fur-ther optimising management.” (line 240-245)

Thank you for your professional comments! Please see the attachment.

 Reference:

  1. Fang QY, Li CB, Lin D. Evaluation of the rationality of perioperative analgesics in 244 cases of orthopedic surgery. Chin J of rural med and pharm 2020;6(27): 39-40. [In Chinese]
  2. Zhu Q, Chen X, Guan JJ. Statistics and rationality evaluation of perioperative analgesic use in orthopedics. J Med Theor & Prac 2020;33(09): 3277-3278. [In Chinese]

Round 2

Reviewer 1 Report

Dear Authors

Thank you for explanations and changes implemented to the manuscript, which seems to be much more appropriate.

Whatsoever I would like to ask you to reconsider two more aspects:

  1. I am aware that experts consensuses you have choosen are excluding the Tramadol from opioids group, but I think it should be commented and clearly highlited, that in many other situations/recommendations etc tramadol is considered as opioid ("weak").
  2. I will be insisting on puting more attention to potential harmfull reactions of drug-drug combinations (with the same mode of action, especially NSAIDs) administered simultaneously. As you are not aware, how the drugs were administered (due to data base limitation) it should be really strongly highlited. You have changed that part of discussion, and it sounds much better, whatsoever I think that you should complement that part.

Author Response

Responses to reviewer 1:

Reviewer #1:

Thank you for explanations and changes implemented to the manuscript, which seems to be much more appropriate. Whatsoever I would like to ask you to reconsider two more aspects:

  1. I am aware that experts’ consensuses you have chosen are excluding the Tramadol from opioids group, but I think it should be commented and clearly highlighted, that in many other situations/recommendations etc. tramadol is considered as opioid ("weak").

Reply 1: Thank you for your important comment.

We have added more explanations about tramadol in the method and discussion section, as well as a comparison of studies on tramadol as a weak opioid, which is enough to emphasize that it is classified as a non-opioid drug in this study to avoid misunderstanding.

Tramadol was classified as a non-opioid central analgesic rather than a weak opioid according to expert consensus in this study.”(line 124-126

In previous studies, tramadol was classified as a weak opioid agonist with two mechanisms of action[1]. And in some study, tramadol was identified as an atypical opioid distinct from opioids[2]. In this study, tramadol was classified as a non-opioid central analgesic based on existing expert consensus in China, which is a reflection of real-world drug utilization method in China and was therefore adopted.(line 210-215)

  1. I will be insisting on putting more attention to potential harmful reactions of drug-drug combinations (with the same mode of action, especially NSAIDs) administered simultaneously. As you are not aware, how the drugs were administered (due to data base limitation) it should be strongly highlighted. You have changed that part of discussion, and it sounds much better, whatsoever I think that you should complement that part.

Reply 2: Thank you for your insightful comment.

We have added more content has been added to highlight the possible hazards of concomitant use of drugs with the same mechanism of action.

“For instance, "celecoxib + parecoxib" was the most common mode of combined drug use, and 29 patients used Sufentanil and Remifentanil at the same time in Mode 3, both of which had the same mechanism of action. Although, there is some evidence showed that if iv parecoxib followed by oral celecoxib in different period after TKA surgery may decrease the use of morphine [3]. However, in a study analyzing international reports of adverse reactions, celecoxib was found to pose a greater risk of adverse reactions when used in combination with other NSAIDs, such as gastrointestinal side effects, renal and urinary disorders, and hypertension [4]. One study also confirmed a higher risk of gastrointestinal bleeding with the combination of different NSAIDs [5]. So, it is still necessary to be vigilant when using drugs with the same mechanism of action during perioperative period and needs further exploration [6].” (line 223-233)

Thank you for your professional comments! Please see the attachment.

Reference:

  1. Scott LJ, Perry CM. Tramadol: a review of its use in perioperative pain. Drugs 2000;60(1):139-76.
  2. Jacob M. Wilson, Andrew M. Schwartz, Kevin X, et al. The impact of preoperative tramadol-only use on outcomes following total knee arthroplasty – Is tramadol different than traditional opioids? The knee 2021; 28:131-138.
  3. Zhuang Q, Bian Y, Wang W, et al. Efficacy and safety of Postoperative Intravenous Parecoxib sodium Followed by ORal CElecoxib (PIPFORCE) post-total knee arthroplasty in patients with osteoarthritis: a study protocol for a multicentre, double-blind, parallel-group trial. BMJ open 2016;6(9): e011732.
  4. Baselyous Y, De Cocinis M, Ibrahim M, Kalra A, Yacoub R, Ahmed R. Potentially inappropriate concomitant medicine use with the selective COX-2 inhibitor celecoxib: Analysis and comparison of spontaneous adverse event reports from Australia, Canada and the USA. Expert Opinion on Drug Safety 2019; 18:3:153-161
  5. Nagata N, Niikura R, Aoki T, et al. Lower GI bleeding risk of nonsteroidal anti-inflammatory drugs and antiplatelet drug use alone and the effect of combined therapy. Gastrointestinal Endoscopy 2014; 80(6):1124-1131.
  6. Richardson J, Holdcroft A. Results of forty years Yellow Card reporting for commonly used perioperative analgesic drugs. Pharmacoepidemiology. Drug Saf 2010;16(6):687-694.

Reviewer 2 Report

The authors substantially approved the manuscript , I suggest that they also include in the abstract  small sentence about  the limitation of the study. 

Author Response

Responses to reviewer 2:

Reviewer #2:

The authors substantially approved the manuscript, I suggest that they also include in the abstract small sentence about the limitation of the study.

Reply: Thank you for your important comment. We have added the sentence of limitations about this study in the abstract (line 31-32) “Due to the limitation of database, this study could not subdivide operation stages, anesthesia methods, dosage forms of drugs.”

Thank you for your professional comments! Please see the attachment.

This manuscript is a resubmission of an earlier submission. The following is a list of the peer review reports and author responses from that submission.

Round 1

Reviewer 1 Report

Chen et al. sought to characterize the types of drugs utilized for pain management during TKA surgery in China. While the paper clearly characterizes these use of these common pain medications, there is not a clear notion as to why this characterization was needed or what impact this characterization has on patient outcomes, cost-to-benefit ratios, or the medical practice. Is there any evidence to suggest that any of these pain medications is superior to the other for pain management, patient outcome, surgical time, etc.? As it is currently written, there are too many unknowns and concerns regarding the implications of such a set of data.

Reviewer 2 Report

This paper wants to speak about analgesia and anesthesia, but without the level of acknowledgment required obviously (anesthetic drugs and analgesic drugs are not even separated in the study). The authors should ask a medical advice and re examination of the study design and the writing.Moreover, the aim and objective of this study are not well defined and clear.

PAGE 2: Although the introduction is very good and detailed, it doesnt allow to understand if this is an economic or a medical study. No primary or secondary outcomes are defined !

PAGE 2 line 94 : no clinicaltrial registration 

P3 l 98: inclusion criterias are not detailed enough 

P3 p109: no mention of tourniquet use

P3 L116: should be detailed with spinal anesthesia, regional anesthesia, etc...

P3 l124: anesthesia and analgesia drugs are all mixed , whereas they are not used at the same moment ! some drugs like sufentanyl/remifentanyl are never used as analgesic durgs bont only for general anesthesia;

P4 l134: more details required 

RESULTS: see previous comments

DISCUSSION: did not read it, as the study is not scientifically strong enough to me, and full of wrong anesthesia/analgesia information

Reviewer 3 Report

In this retrospective study the Authors aimed to investigate the drug utilization for pain management during perioperative period of total knee arthroplasty using real-world data. The study is interesting, however there are some minor issues to be addressed.

My specific comments are as follows:

- The Authors should specify the modality of analgesic drugs administration (e.g.: oral, periarticular infiltration…)

- 3.1.1 “Characteristics of patients” and 3.1.2 “Surgical characteristics:  The Authors should not repeat information already reported respectively in the table 1 and in the table 2.

- Discussion: The Authors did not discuss the results concerning the association between drugs and surgery types (unilateral vs bilateral) e anesthesia method (local vs general). These results should be commented in the discussion.

Reviewer 4 Report

Major comment –

  1. Multicentric retrospective study on analgesics use in TKA patients. Overall study was designed and analyzed well.
  2. What are the ultimate aims and objectives of this study?
  3. Authors did not look at the level of pain control – both subjective (pain scores) as well as objective (total dosages of analgesic requirement)
  4. Complications if any in any of the analgesic groups? If opioid containing groups have higher incidences of complications or NSAIDS group has shown lack of side effects that can justify using more expensive drugs from NSAIDS group

Minor comments –

  1. L46, P2 – Please simplify the definition of TKA
  2. L52,53, P2 – are repetitive – L52 can be deleted.
  3. Please explain what was meant by local anesthesia – in general, neuraxial anesthesia is the standardized term for spinal or epidural anesthesia, peripheral nerve block is used for when a particular peripheral nerve is blocked by administering local anesthetics. Local anesthesia refers to local anesthetic infiltration by surgeons.
  4. Patients who underwent GA, required maximum percentage of mode 1 analgesics (opioid containing) and least percentage of Mode 2 (only NSAIDS) analgesics. Does that mean Local anesthesia provides additional analgesia that lasted in the postoperative period and impacted analgesic regimen
  5. L192 – Rephrase the heading to ‘Cost analysis of analgesic drugs’
  6. L193 – Rephrase the sentence as ‘the costliest …’
  7. L204 – Change to ‘multimodal analgesia’ instead of ‘multimodal analgesic’ in the sentence that starts with “in this retrospective study …’
  8. L208 – Rephrase the sentence with ‘In our study, the NSAIDS were most commonly prescribed group of analgesics – this finding was in accordance with previous studies quoted in literature’
  9. L241 – It was nice of authors to explain how details about surgery can affect pain management. More complex surgery has potential to be more painful